# Optimized Traffic Light System with AIC and Application to the 2021 M6.7 Yangbi Earthquake Sequence

**DOI:** 10.3390/e25050759

**Published:** 2023-05-06

**Authors:** Rui Wang, Ying Chang, Peng Han, Miao Miao, Zhiyi Zeng, Haixia Shi, Danning Li, Lifang Liu, Youjin Su

**Affiliations:** 1Department of Earth and Space Science, Southern University of Science and Technology, Shenzhen 518055, China; 11930870@mail.sustech.edu.cn (R.W.); hanp@sustech.edu.cn (P.H.); miaom@sustech.edu.cn (M.M.); 11930855@mail.sustech.edu.cn (Z.Z.); 2Institute of Mining Engineering, BGRIMM Technology Group, Beijing 100160, China; 3Guangdong Provincial Key Laboratory of Geophysical High-resolution Imaging Technology, Southern University of Science and Technology, Shenzhen 518055, China; 4China Earthquake Networks Center, Beijing 100045, China; shihaixia08@seis.ac.cn; 5Earthquake Administration of Yunnan Province, Kunming 650224, China; zuni_2001@163.com (D.L.); lifang_l@sina.com (L.L.); suyoujin0818@sina.com (Y.S.)

**Keywords:** seismicity, traffic light system, *b*-value, AIC, Yangbi earthquake

## Abstract

One important question in earthquake prediction is whether a moderate or large earthquake will be followed by an even bigger one. Through temporal *b*-value evolution analysis, the traffic light system can be used to estimate if an earthquake is a foreshock. However, the traffic light system does not take into account the uncertainty of *b*-values when they constitute a criterion. In this study, we propose an optimization of the traffic light system with the Akaike Information Criterion (AIC) and bootstrap. The traffic light signals are controlled by the significance level of the difference in *b*-value between the sample and the background rather than an arbitrary constant. We applied the optimized traffic light system to the 2021 Yangbi earthquake sequence, which could be explicitly recognized as foreshock–mainshock–aftershock using the temporal and spatial variations in *b*-values. In addition, we used a new statistical parameter related to the distance between earthquakes to track earthquake nucleation features. We also confirmed that the optimized traffic light system works on a high-resolution catalog that includes small-magnitude earthquakes. The comprehensive consideration of *b*-value, significance probability, and seismic clustering might improve the reliability of earthquake risk judgment.

## 1. Introduction

The Gutenberg–Richter (G-R) law, logN=a−bM (where *N* is the cumulative number of earthquakes above magnitude *M*, *a* is the intercept presenting the productivity and *b* is the slope), describes magnitude–frequency distribution [1,2]. The *b*-value has been proved to be inversely correlated with underground differential stress levels through laboratory experiments and seismic studies [3,4,5,6,7]. Hence, decreasing *b*-value or low *b*-value indicate higher earthquake risk. To date, *b*-value has been widely applied in seismic studies [8,9,10,11,12,13,14,15,16,17,18]. It has been reported that large earthquakes tend to occur in regions with low *b*-values [19,20], and a temporal decrease in *b*-values around the epicenter can indicate a larger upcoming earthquake [10,21,22]. Therefore, the *b*-value is often used for medium- to long-term earthquake predictions.

Foreshocks are one of the most effective precursors for short-term earthquake prediction. The spatial-temporal evolution of foreshocks can reflect the state of stress and strength near the mainshock, which can be represented by *b*-value [23]. To identify whether a moderate to large earthquake is likely to be followed by an equally large or larger one, Gulia and Wiemer proposed a traffic light system that compares the *b*-value after the earthquake with the background *b*-value (before the earthquake) [24]. They set a threshold value of 10%, meaning that if the *b*-value decreases by more than 10%, the region should be considered high risk, with a high probability of a major earthquake. This system has been tested in multiple earthquake cases, but the criteria are subjective, and are not able to provide the significance of the *b*-value difference and reliability of the risk estimate. We used a method based on the Akaike Information Criterion (AIC) and bootstrap with the aim of reducing false positives and ensuring effectiveness, allowing us to determine the probability of significant differences in *b*-value [18,25,26]. The 2021 Yangbi earthquake sequence was scanned using the optimized traffic light system to identify the mainshock.

On 18 May 2021, an M4.7 (Mw4.2) earthquake occurred in Yangbi County, Yunnan Province, China, followed by a series of earthquakes, which included an M4.7 earthquake on 19 May and an M5.9 earthquake on 21 May. These earthquakes were foreshocks of the mainshock, a magnitude 6.7 (Mw6.1) earthquake that occurred on 21 May (Figure 1). As a typical foreshock–mainshock–aftershock sequence and the first M > 6.0 earthquake in Yunnan province since 6 December 2014, the 2021 Yangbi earthquake sequence has been widely studied [27,28,29,30,31]. In this study, we analyze the temporal *b*-value change using the earthquake catalog from China Earthquake Networks Center (CENC) between 1 January 2020, and 31 July 2021. Then, we apply the traffic light system and give the probability of significance through AIC and bootstrap [25,32]. The spatial *b*-value before the 18 May 2021 M4.7 earthquake, the 21 May 2021 M6.7 earthquake, and 31 July 2021 are obtained using the Hierarchical Space–Time Point–Process Models (HIST-PPM) proposed by Ogata [33]. The *b*-value differences are computed to reveal possible features of foreshock and aftershock. Zhou et al. provided a high-resolution seismic catalog using an AI picker and a matched filter [30]. In the discussion, we combine the two catalogs into a new catalog and further validate our results using the new catalog, except the method of estimating spatial *b*-value replacing with the grid search method. In addition, we analyze the seismic clustering of the Yangbi earthquake sequence on the basis of a new parameter. Then, we discuss the effect of appending the earthquake catalog to the two spatial *b*-value methods and the feasibility and performance of long- and short-term seismic risk estimates. Finally, we discuss the influence of the choice of study area on the results. Our results show that the new method can be used to determine the significance of the change in *b*-value after the foreshock. In addition, by studying the Yangbi earthquake sequence, it is confirmed that, for foreshock–mainshock–aftershock sequences, an analysis of the *b*-value of the foreshock can be used to predict the mainshock.

## 2. Materials and Methods

### 2.1. Data

The 2021 Yangbi earthquake sequence occurred near the southern part of the Qiaohou–Weishan fault, and formed a seismic zone referred to as the Yangbi seismic zone by Lei et al. [29]. In this study, as shown in Figure 1, we chose Yangbi seismic zone and its southeast extension as study area 1, while spatial *b*-values were calculated in the area between 99.0° E−101.0° E and 24.5° E−26.5° N, including study area 1, using HIST-PPM. The earthquake catalog used in this study was obtained from China Earthquake Networks Center (CENC).

### 2.2. Conventional Method for Determining b-Value

This study began with the CENC earthquake catalog, and maximum curvature (MAXC) was applied to estimate the magnitude of completeness ($M_{c}$), and maximum likelihood estimation (MLE) was used in combination with bootstrap to estimate the *b*-value and the error of the *b*-values [26,35,36]. Temporal analysis of the *b*-values was performed for the period from 1 January 2020 to 31 July 2021. Each sample window included 300 seismic events, and moved forward chronologically with a step size of 1 event. For each sample window, the *Mc* was estimated using the MAXC method, and earthquakes equal to or greater than the threshold magnitude (*Mc* + 0.1) were used to compute the *b*-value using MLE.

Based on the G-R law, the frequency of earthquake magnitude was defined in terms of the condition intensity function, as follows:(1)$$\lambda \left(M\right)=10^{a-bM}=Ae^{-\beta M}$$
where $M>M_{c}$ and β=bln10. The probability density distribution of magnitude can be derived as follows:
*f*(*M*) = *βe* − *β*(*M* − *M_c_*)(2)Aki proposed a likelihood function (Aki, 1965), as follows:(3)$$L\left(\beta \right)=\prod_{i=1}^{n}\beta e^{-\beta \left(M_{i}-M_{c}\right)}$$Using the maximum likelihood function, the *b*-value can be estimated as follows:(4)$$b=\frac{1}{\ln\left(10\right)\left(\bar{M}-M_{c}\right)}$$
where $\bar{M}$ is the mean magnitude of events in the sample window. Bootstrapping was repeated 1000 times and 1000 $M_{c}$ and 1000 *b*-values were obtained for each sample window, the mean values of which were adopted as the $M_{c}$ and *b*-values, respectively, of the sample window. In addition, the time of the last event in the sample window was taken as the time of the $M_{c}$ and *b*-value, and all results measure after 1 January 2021 were kept, as shown in Figure 2a–c.

For the spatial *b*-value, this study employs two methods. One of them, applied in the discussion in this study, is to establish a grid in study area 1, searching for earthquake events within a certain radius or a certain closet number, centering the grid point as the sample window and the *b*-value and error of the point are obtained using MAXC and MLE in combination with bootstrap [37]. The significance level of the difference between *b*-values determined using MLE in two sample windows can be estimated using ΔAIC [25], as follows:(5)$$\Delta AIC=-2\left(N_{1}+N_{2}\right)\ln\left(N_{1}+N_{2}\right)+2N_{1}\ln\left(N_{1}+\frac{N_{2}b_{1}}{b_{2}}\right)+2N_{2}\ln\left(N_{2}+\frac{N_{1}b_{2}}{b_{1}}\right)-2$$
where $N_{1}$ and $N_{2}$ are the numbers of earthquakes occurring in the two sample windows, and $b_{1}$ and $b_{2}$ are the *b*-values of the two sample windows, respectively. The probability of there being no difference between the two sample windows can be expressed as follows [38]:(6)$$P_{b}=\exp\left(-\frac{\Delta AIC}{2}-2\right)$$

### 2.3. HIST-PPM

In light of the fact that the *β* value in Equation (3) is dependent on location and/or time, Ogata proposed the Hierarchical Space–Time Point–Process Models (HIST-PPM) [33]. In HIST-PPM, it is assumed that *β* is a function of the epicenter $\left(x_{i},y_{i}\right)$, and since *β* is positive, the parameterized $\beta \left(x_{i},y_{i}\right)$ can be determined as follows:(7)$$\beta =\beta \left(x_{i},y_{i}\right)=e^{\phi_{\theta }\left(x_{i},y_{i}\right)}$$
where the $\phi_{\theta }$ is the 2D *B*-spline function and θ is the coefficient of the function $\phi_{\theta }$. In HIST-PPM, the study area is tessellated by the Delaunay triangle with the epicenter as the apex [14,33]. Then, to avoid spatial mutation and overfitting of results, instead of the maximum likelihood function method, Ogata estimated the parameter θ by maximizing the penalized log-likelihood as follows:(8)$$R\left(\theta |w\right)=\ln L\left(\theta \right)-Q(\theta |w)$$
The Q(θ|w) is the penalty term correlated with spatial derivation, defined as:(9)$$Q\left(\theta |w\right)=w\iint \left\{{\left(\frac{\partial \phi_{\theta }\left(x,y\right)}{\partial x}\right)}^{2}+{\left(\frac{\partial \phi_{\theta }\left(x,y\right)}{\partial y}\right)}^{2}\right\}dxdy$$
where w is the weight optimized by Akaike’s Bayesian Information Criterion (ABIC) [14,33]. Accessing the *b*-values on the apexes, the values in the triangle can be obtained by interpolation at the 0.005° × 0.005° node.

### 2.4. Foreshock Identification

Gulia and Wiemer introduced a traffic light system for earthquake hazard assessment [24]. They first determined the median value of the *b*-value before the occurrence of a foreshock and used it as the background value. Then, they calculated the percentage difference in the *b*-value after a moderate-to-large earthquake. If the percentage decrease in *b*-value was more than 10%, it was flagged as red, indicating a high risk of earthquake hazard. If the percentage increase in *b*-value was more than 10%, it was tagged as green, representing a low risk. If the change in *b*-value was between these two thresholds, it was tagged as yellow, indicating an undetermined risk.

The AIC method in combination with bootstrap was applied by Xie et al. on the Pacific Coast of Tokachi, Hokkaido, Japan [18]. In this method, first, a temporal *b*-value of the study sequence is obtained by MLE in combination with bootstrap. In this study, the earthquakes from 1 January 2020 to 18 May 2021, with magnitudes greater than M4.7, were chosen as samples for background reference, and 1000 times bootstrap was applied with a sample size of 300, and the same with the sample window. Then, the 1000 ΔAICs of 1000 bootstrap samples with a sample window and the percentage of ΔAIC>2 (P(ΔAIC>2)) are obtained, where P(ΔAIC>2) indicates the probability of a significant difference occurring between the background reference and the sample window.

## 3. Results

Figure 2a shows the earthquakes in the study area, including the Yangbi earthquake sequence. The $M_{c}$ analysis in Figure 2b shows that before the Yangbi earthquake, $M_{c}$ was about 1.0. Small fluctuations can be observed between the M4.7 foreshock and the mainshock. Immediately after the mainshock, $M_{c}$ reaches a short-term peak, before fluctuating and stabilizing at 0.6. As shown in Figure 2c,d, temporal variation in the *b*-value was previously stable at 1.0. Immediately after the M4.7 foreshock, the *b*-value decreased by more than 10% to 0.7, indicating high risk. Then, *b*-value increased to 1.0 immediately after the mainshock, rolling into the yellow region and up to the green region, indicating low risk, before finally stabilizing between the red and yellow regions. The temporal variation in *b*-value shows the characteristics of a foreshock–mainshock–aftershock sequence, and can be recognized by the traffic light system. As shown in Figure 2e, which depicts the P(ΔAIC>2) between the sample window and background reference, a low P(ΔAIC>2) can be observed before the M4.7 foreshock. Immediately after the M4.7 foreshock, the P(ΔAIC>2) reaches 40% and increases to more than 80%, remaining high until the occurrence of the mainshock. After the mainshock, P(ΔAIC>2) declines rapidly, peaking at about 60% and fluctuating slightly before stabilizing at 20%. Subsequently, P(ΔAIC>2) increased to 50% at the end of June. Compared with P(ΔAIC>2), the probability of P(ΔAIC>2&Δb<0) indicates the significance of a decrease in *b*-values, and presents a relatively high value, matching *b*-values that are lower than the background, especially during the foreshocks. The difference between P(ΔAIC>2) and P(ΔAIC>2&Δb<0) is P(ΔAIC>2&Δb≥0), indicating the significance of an increase in the *b*-value, which reached relatively high values at the end of June 2021.

Figure 3a–c display the spatial distribution of *b*-values. Generally, low *b*-values are observed in the vicinity of the mainshock. Figure 3d illustrates the difference in *b*-value between Figure 3a,b, indicating that foreshocks lead to a decrease in *b*-values around the epicenter. The difference in *b*-values between Figure 3b,c in Figure 3e suggests that aftershocks cause an increase in *b*-value around the epicenter and a disruption in *b*-values nearby.

## 4. Discussion

A temporal analysis of the *b*-values in the Yangbi earthquake sequence indicates that the *b*-value decreased immediately after the M4.7 foreshock and increased after the mainshock. This pattern matched the traffic light system and could be tested using it. P(ΔAIC>2,Δb<0) provides information on the significance of the decrease in *b*-value compared to the background and helps to determine the risk level. After the M4.7 foreshock, the decrease in *b*-value was significant, indicating a high risk level. However, after the mainshock, P(ΔAIC>2,Δb<0) fell back, indicating a relatively low risk. P(ΔAIC>2,Δb<0) can provide the significance of the decrease in *b*-value compared to the background and give the risk level. A threshold=μ+5σ (with 99.99994% confidence) of P(ΔAIC>2,Δb<0) can also be set, where *μ* and *σ* are the mean and standard deviation of P(ΔAIC>2,Δb<0) in the background. If the *b*-value decreases and P(ΔAIC>2,Δb<0)> threshold, this may indicate that an equal or larger earthquake may follow.

As a statistical method, *b*-value results can be affected by the quality of the earthquake catalog. To obtain a more elaborate evolution of *b*-values and to verify whether the results are dependent on the quality of the earthquake catalog, a higher-quality earthquake catalog is necessary. We combined the catalog of Zhou et al., a high-resolution seismic catalog produced using an AI picker and a matched filter, with that of CENC [39]. Due to the lack of conformity in the magnitude reported by the two catalogs, we performed a first- and second-order linear regression between the same events in the catalogs of CENC and Zhou et al. (shown in Appendix A), and selected the first-order linear regression coefficients shown in Equation (10) to convert the magnitudes reported in the Zhou et al. catalog. The results are shown in Appendix A.
(10)$$M\left(CENC\right)=1.0779\times M\left(Zhou\right)-0.16182$$

A temporal analysis of the newly combined catalog is presented in Figure 4. Which shows that the trend of $M_{c}$ remains mostly unchanged. In Figure 4c, the *b*-value decreases into the red region, indicating a high risk level, immediately after the M4.7 foreshock and increases into the yellow region after the mainshock, which is similar to the results in Figure 2c, and can also be tested using the traffic light system, verifying the reliability of the results. Comparing the *b*-values in Figure 2c and Figure 4, there is a slight increase in the *b*-value immediately before the mainshock and a more significant rise after the mainshock in Figure 4. Based on this, it can be inferred that there is a low risk of earthquake after the mainshock. Therefore, the inclusion of the Zhou et al. [39] catalog helps to estimate the risk after the mainshock.

As shown in Figure 5a–c, the spatial evolution of the *b*-value in the new catalog presents the same features as the temporal variation in *b*-values presented in Figure 2c and Figure 4c and the spatial evolution of *b*-values presented in Figure 3a–c. *b*-values around the mainshock epicenter decreased when the foreshock occurred and increased after the mainshock. The significances of the variations in *b*-value are presented in Figure 5d,e demonstrating that the results are reliable.

In this study, two methods were used to obtain spatial *b*-values: HIST-PPM and the grid search method. HIST-PPM provides high-spatial-resolution results, but it does not estimate the significance of difference in *b*-value. Moreover, it may not provide enough information to analyze the new catalog, which combines earthquakes of smaller magnitude. To ensure the completeness of samples, a higher $M_{c}$ should be set for earthquakes before the Yangbi earthquake, as shown in Figure 2b and Figure 4b. The contribution of the catalog of Zhou et al. is that it mainly focuses on small earthquake events, which may be truncated when using HIST-PPM. In contrast, in the grid search method, the samples at each grid point are analyzed separately, and the significance of differences in *b*-value are determined. Therefore, in this study, the HIST-PPM method was used to obtain high-spatial-resolution results, while the grid search method was used to verify the results’ reliability.

Earthquake clustering, or seismic concentration, is a significant indicator of critical stress state underground. Lippiello et al. proposed the use of the quantity $\phi =R^{-1}/R_{b}{}^{-1}$ to evaluate the concentration ratio of seismicity, where $R^{-1}$ is the average inverse distance from the mainshock source to n events that occur before a given time, and $R_{b}{}^{-1}$ is the same as $R^{-1}$ of n events before the first event of $R^{-1}$ [40]. ϕ>1 indicates concentrated seismicity and ϕ<1 indicates dispersive seismicity. The ϕ variant $D={\bar{R}}^{-1}/{\bar{R_{b}}}^{-1}$ is applied in this study. The inverse of the average distance between events in each pair of the n events before a given time is ${\bar{R}}^{-1}$, and ${\bar{R_{b}}}^{-1}$ is the same as ${\bar{R}}^{-1}$ of n events before the first event of ${\bar{R}}^{-1}$. The meaning of D is the same as ϕ, while the difference is that D can be used in real-time analysis without the mainshock location.

In this study, the method is applied to two catalogs, with $M_{c}=1.2$ and a window length n=25. The resulting D values are shown in Figure 6. The error of D is estimated using 100 bootstrap samples. Prior to the M4.7 foreshock, D was approximately 1, indicating no significant clustering or dispersion of seismicity. However, around March 1, 2021, D exceeded 1, indicating the presence of small-scale clustering. After the M4.7 foreshock, D increased significantly to 15 in Figure 6a and 40 in Figure 6b, indicating the occurrence of a large-scale cluster of earthquakes. Subsequently, D exhibited dramatic increases and decreases, suggesting complex seismicity resulting from the complex stress evolution caused by the source rupture.

Earthquake clusters are a kind of seismicity wherein earthquakes occur intensively in time and space. In many cases, clusters are earthquakes triggered on a large scale, in an unstable stress state. There are always co-seismic and post-seismic clusters because of the stress adjustment caused by the earthquake rupture. On the one hand, a moderate earthquake followed by a relative large-scale cluster sends a warning that the area is in a critical condition. On the other hand, a significant decrease in *b*-value always indicates a high risk of moderate–large earthquakes. Therefore, clustering along with a significant decrease in *b*-value would be a better indicator of seismic risk for short-term follow-up. For short-term earthquake prediction, high-density and multiclass observations are necessary [41]. Thus, observations should be targeted and focused on key areas. Prior research has reported mid- to long-term earthquake predictions that were effective for Yangbi earthquakes, providing a reference for the selection of key areas [12]. A better solution would be to combine mid- to long-term and short-term earthquake predictions. By conducting a large-scale scan of the spatial evolution of the *b*-value, a reference for areas at high risk of large earthquakes can be accessed for mid- to long-term prediction. High-density and multiclass stations can then be set up in high-risk areas to proceed with real-time observations. The combination of *b*-value, probability of significance, and *D* can provide effective earthquake prediction. In addition, Rundle et al. proposed an earthquake nowcasting method based on the computation of the earthquake potential score (EPS) [42], and a recent study in Greece suggested that it was useful [43]. Integrating information from different approaches and observations might be a possibility for improving the efficiency of earthquake forecasting [44,45,46].

We also investigated the influence of target region selection. We chose a smaller study area, as shown in Appendix A. The temporal variation of the earthquakes in the region shown in Appendix A is presented in Appendix A. The trends of the *b*-value in the new study area are similar to those shown in Figure 2c and Figure 4c. The major characteristic, whereby the *b*-value decreased immediately after the M4.7 foreshock and increased after the mainshock, is also distinct, which proves that the features of the results in Figure 2c and Figure 4c are reliable and independent of the adjustment of the study area.

The accurate estimation of $M_{c}$ is crucial for seismicity studies, as an underestimation of $M_{c}$ can result in a misestimation of the *b*-value [35]. For reliability testing, we also utilized the normalized distance test method (NDT), which combines the normalized K-S distance test with bootstrap resampling, in order to achieve more accurate estimations of $M_{c}$ and *b*-value [47]. As shown in Figure 7, the $M_{c}$ estimated by the NDT method is about 0.5 higher than that estimated by the MAXC method, which is consistent with the findings of Lombardi (2021). The *b*-value estimated using the NDT method is consistent with that obtained using the MAXC and MLE methods presented in Figure 2.

To test the reliability of our results, we added 0.5 to the original $M_{c}$ in Figure 2b, as suggested by the estimation of $M_{c}$ obtained using NDT, and MLE was applied to obtain the *b*-value. We then analyzed the significance of the change in *b*-value and earthquake clusters, as shown in Figure 8. The decrease in *b*-value after the M4.7 foreshock reached 10%. The significance probability increased by more than 5 times the error, indicating a significant decrease in *b*-value. The significance probability reported in Figure 8b is lower than the significance probability presented in Figure 2e, because higher magnitude thresholds result in smaller sample sizes. The D of the $M_{c}+0.5$ constrained earthquakes reached 8 between the M4.7 foreshock and the M6.7 mainshock, indicating the occurrence of a large-scale cluster of earthquakes. By employing both the NDT method and the MLE method with a threshold of $M_{c}+0.5$ to obtain the $M_{c}$ and *b*-value, we found a consistent temporal variation in *b*-values indicating the significance probability. Therefore, the optimized traffic light system was verified to be effective.

Most previous *b*-value estimations have been influenced by the incompleteness of the earthquake catalog. In order to address this issue, Nicholas (2021) proposed a novel method called “B-positive” [48]. Based on the statistics of difference in magnitude between successive earthquakes ($M^{\prime }$), this method is designed to be insensitive to transient changes in catalog completeness. We apply the B-positive method to the Yangbi earthquake sequence and compare the results with those presented in Figure 2c, as shown in Figure 9. The estimation of *b*-value using the B-positive method shows a similar trend to the results obtained using the MLE method (the purple line in Figure 9b). These temporal *b*-values may have different values, but this does not affect the application of our optimized traffic light system based on changes in *b*-value. Comprehensive analysis of *b*-value using multiple methods can improve the reliability of the results and promote better application of our method.

## 5. Conclusions

The temporal and spatial variations in *b*-values before and after the Yangbi earthquake were calculated. An improved traffic light system employing AIC and bootstrap analysis was applied spatially and temporally to the Yangbi earthquake sequence and was demonstrated to possess good efficiency for recognizing foreshock–mainshock–aftershock when used with an appended AI catalog. The AIC method in combination with bootstrap can be used to determine the significance of the difference between foreshocks and the background. In the Yangbi earthquake sequence, the *b*-value significantly decreased immediately after the M4.7 foreshock and increased after the mainshock, which can be tested using the traffic light system in both time and space. The significance probability represents a method for identifying the evolution of *b*-values in both time and space. The seismic clustering parameter also indicates the mainshock following the M4.7 foreshock. Comprehensive consideration of *b*-values, significance probability, and seismic clustering can improve the reliability of earthquake risk judgment.

## Figures and Tables

**Figure 1 entropy-25-00759-f001:**
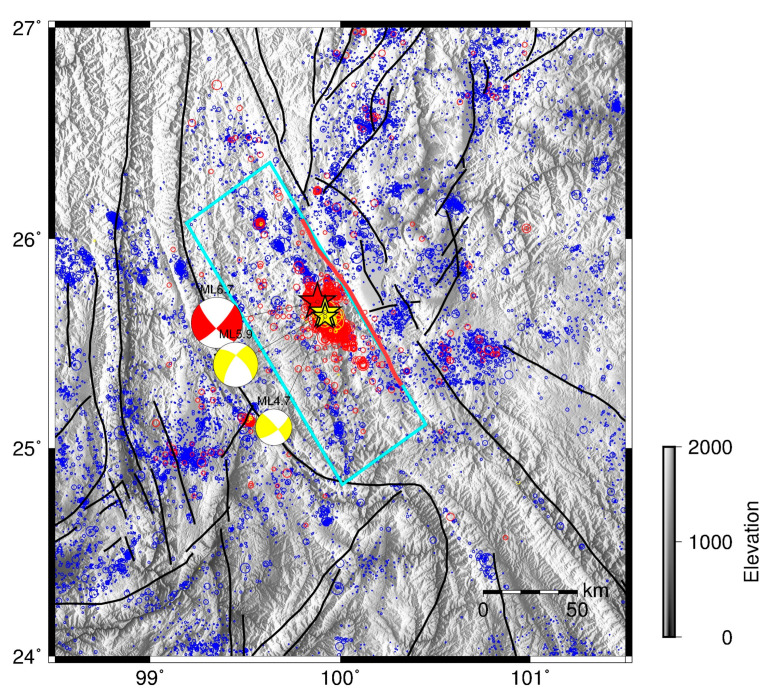
The earthquake distribution around the Yangbi earthquake and the Yangbi sequence. The black solid lines show the main faults in this area [34]. The bold solid red line shows the south part of the Qiaohou–Weishan fault. The light blue line encloses the study area for the spatial *b*-value predicted using HIST-PPM. The blue circles represent the earthquakes from 1 January 2017 to 18 May 2021 M4.7 foreshock. The red circles represent the earthquakes after the M6.7 Yangbi earthquake. The yellow circles represent the earthquakes between the M4.7 foreshock and the M6.7 Yangbi earthquake. The red star shows the epicenter of the M6.7 mainshock. The yellow stars show the epicenters of the M4.7 foreshock and the largest M5.9 foreshock. The corresponding focal mechanisms are labeled.

**Figure 2 entropy-25-00759-f002:**
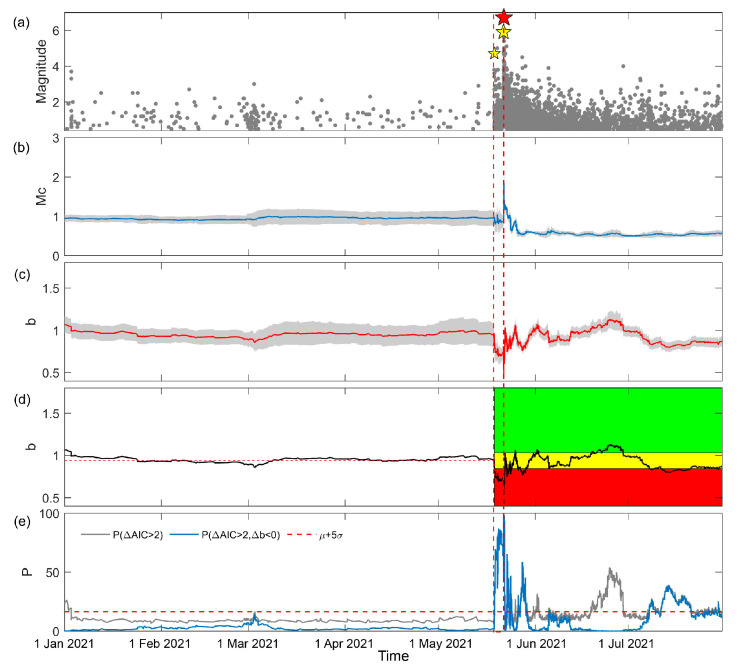
Temporal *b*-value analysis of the Yangbi earthquake sequence. (**a**) The magnitude–time of earthquakes in the study area shown in Figure 1 between 1 January 2021 and 1 July 2021. The red star is the mainshock. The yellow stars are the M4.7 foreshock and the largest M5.9 foreshock. (**b**) The temporal variation in $M_{c}$ with a window length of 300 and a step length of 1. (**c**) The temporal variation in *b*-values with window length of 300 and step length of 1. The gray shading in (**b**,**c**) represents the uncertainty determined through bootstrap. (**d**) Application of the traffic light system to the temporal variation in *b*-values. The black line is the temporal variation in *b*-values. The red horizontal dashed line is the median background level of *b*-values before the occurrence of the M4.7 foreshock. The red, yellow, and green shading represent *b*-values decreasing by more than 10%, increasing by more than 10%, and changing by −10~+10%. (**e**) The probability of significance. The gray line is the percentage of ΔAIC>2. The blue line is the percentage of ΔAIC>2 and Δb<0. The red horizontal dashed line is the mean and five times the standard deviation of the background P(ΔAIC>2,b<0). The red dashed lines through (**a**–**e**) indicate the time-horizon of the foreshock.

**Figure 3 entropy-25-00759-f003:**
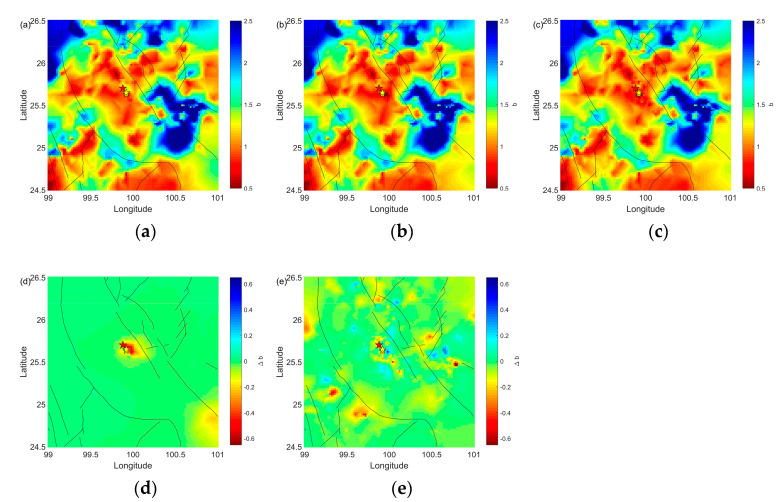
Spatial evolution of *b*-value with $M_{c}\;$= 1.5, and the difference between them. The red star shows the epicenter of mainshock of the Yangbi earthquakes. The study area is divided into a 0.005° × 0.005° grid. The yellows stars show the epicenters of M4.7 foreshock and M5.9 foreshock. (**a**) Spatial *b*-value from 1 January 2017 until the 18 May 2021 M4.7 foreshock. (**b**) Spatial *b*-value from 1 January 2017 until the 21 May 2021 M6.7 mainshock. (**c**) Spatial *b*-value from 1 January 2017 to 1 July 2021. (**d**) Difference between (**a**) and (**b**) ((**b**) – (**a**)). (**e**) Difference between (**b**) and (**c**) ((**c**) – (**b**)). For (**a**–**c**), red in the color bar represents low *b*-values. For (**d**,**e**), red in the color bar represents decreasing *b*-values.

**Figure 4 entropy-25-00759-f004:**
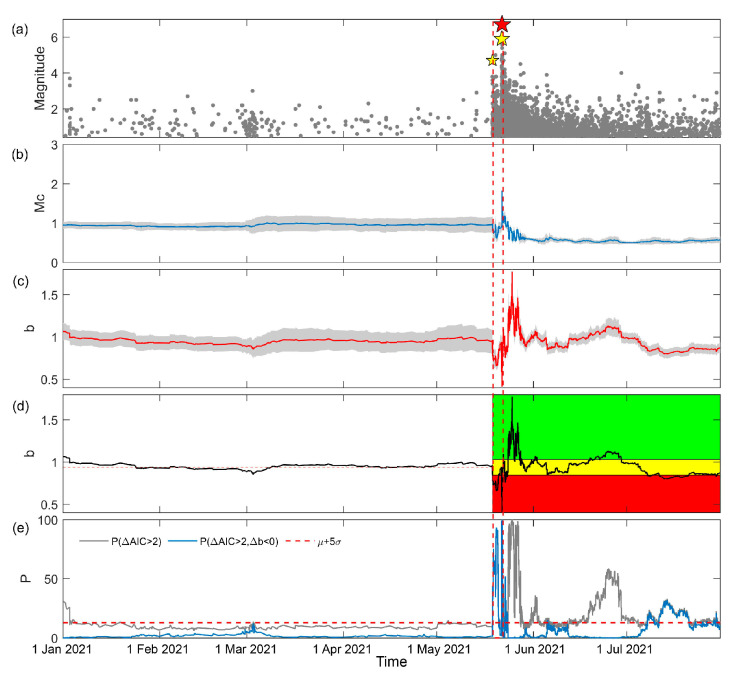
Temporal *b*-value analysis of the new combined catalog of the Yangbi earthquake sequence. (**a**) The magnitude–time of the earthquakes in the new catalog for the study area shown in Figure 1 from 1 January 2021 to 1 July 2021. The red star represents the mainshock. The yellow stars are the M4.7 foreshock and the largest M5.9 foreshock. (**b**) The temporal variation in $M_{c}$ with a window length of 300 and a step length of 1. (**c**) The temporal variation in *b*-value with a window length of 300 and a step length of 1. The gray shading in (**b**,**c**) represents the uncertainty of the results obtained through bootstrap. (**d**) The traffic light system applied on temporal variation in *b*-value. The black line represents the temporal variation in *b*-value. The red horizontal dashed line is the median of *b*-value before the M4.7 foreshock presents background level. The red, yellow, and green shadings denote *b*-value decreasing by more than 10%, increasing by more than 10% and changing by −10~+10%. (**e**) Probability of significance. The gray line is the percentage of ΔAIC>2. The blue line is the percentage of ΔAIC>2 and Δb<0. The red horizontal dashed line is mean and five times the standard deviation of the background P(ΔAIC>2,Δb<0). The red dashed lines through (**a**–**e**) indicate the time-horizon of the foreshock.

**Figure 5 entropy-25-00759-f005:**
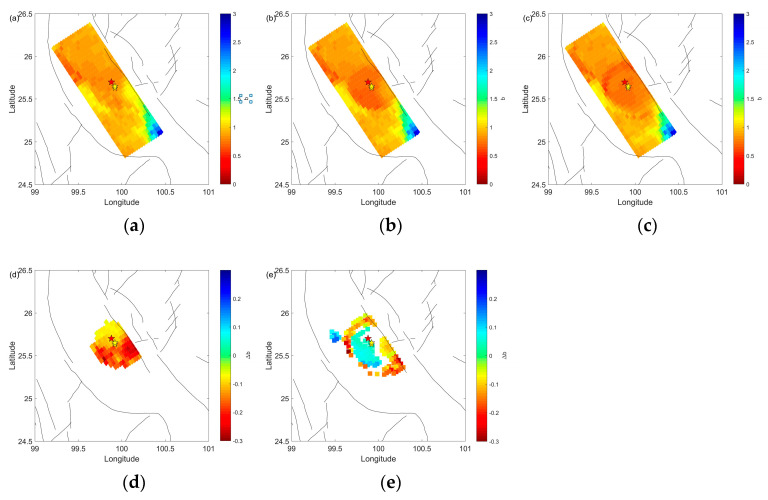
Spatial evolution of the *b*-value in the new catalog and the difference between them. The red star shows the epicenter of the mainshock of the Yangbi earthquake sequence. The yellows stars show the epicenters of the M4.7 foreshock and the M5.9 foreshock. The sampling radius was 30 km and the minimum sample size was 50. (**a**) Spatial evolution of the *b*-value from 1 January 2017 until the 18 May 2021 M4.7 foreshock. (**b**) Spatial evolution of the *b*-value from 1 January 2017 until the 21 May 2021 M6.7 mainshock. (**c**) Spatial evolution of the *b*-value from 1 January 2017 to 1 July 2021. (**d**) Significant difference between (**a**) and (**b**) ((**b**) − (**a**)). (**e**) Significant difference between (**b**) and (**c**) ((**c**) − (**b**)). For (**a**–**c**), red in the color bar represents low *b*-values. For (**d**,**e**), red in the color bar represents decreasing *b*-values.

**Figure 6 entropy-25-00759-f006:**
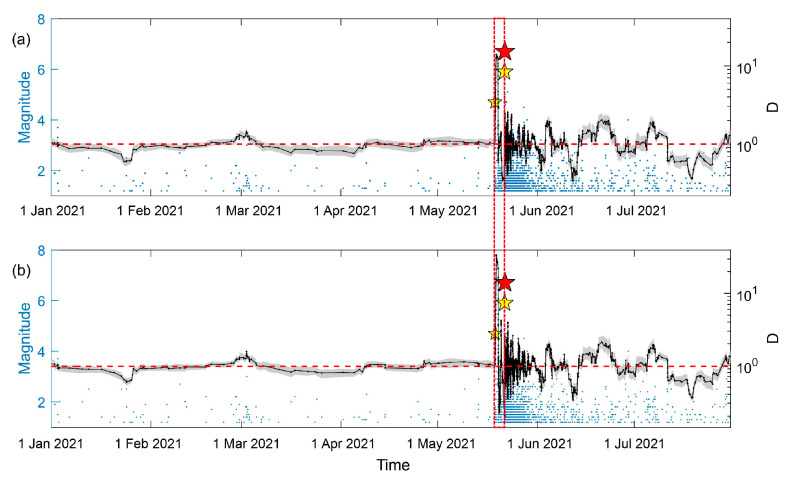
Temporal variation in *D* and earthquake magnitudes. The black solid line represents *D*. Black points highlight the *D* value when earthquakes occur. Gray shading represents the error of *D*. Blue points indicate earthquakes. The red star represents the mainshock of the Yangbi earthquakes. The yellow stars are the M4.7 foreshock and the M5.9 foreshock. (**a**) Temporal variation in *D* using catalog of CENC. (**b**) Temporal variation in *D* using the new catalog.

**Figure 7 entropy-25-00759-f007:**
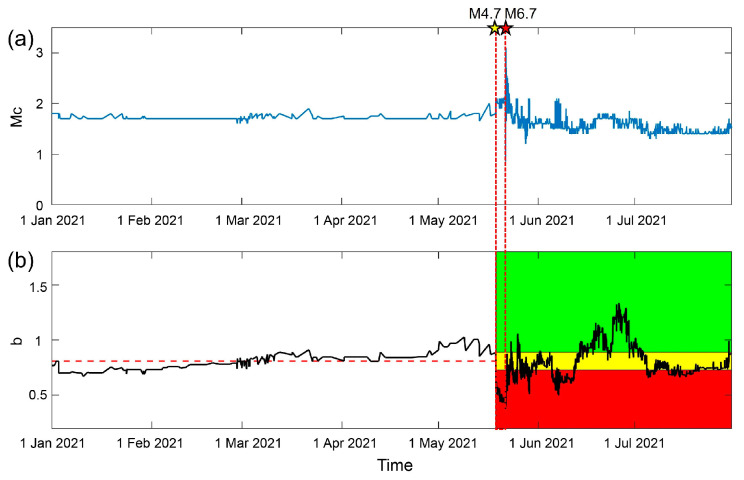
Analysis of the temporal evolution of $M_{c}$ and *b*-value for the Yangbi earthquake sequence using the NDT method. (**a**) Temporal evolution of $M_{c}$ determined using the NDT method with a window length of 300 and a step length of 1. (**b**) Application of the traffic light system to the temporal variation in *b*-value. The black line is the temporal variation in *b*-value. The red horizontal dashed line is the median background level of *b*-values before the M4.7 foreshock occurred. The red, yellow, and green shading correspond to *b*-values decreasing by more than 10%, increasing by more than 10%, and changing by −10~+10%. The red dashed lines through (**a**,**b**) indicate the time-horizon of the foreshock.

**Figure 8 entropy-25-00759-f008:**
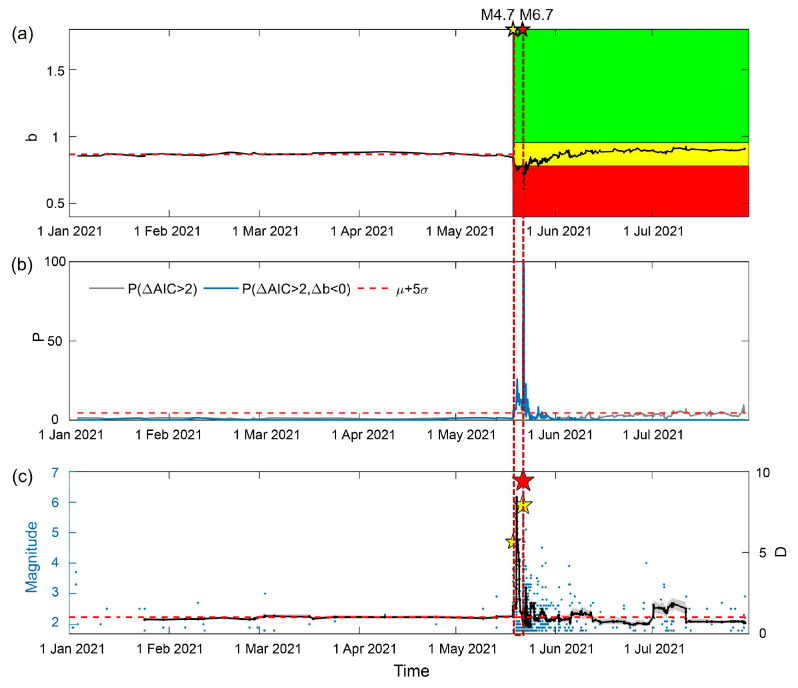
The temporal evolution of the *b*-value, significance, and earthquake cluster analysis with $M_{c}$ in Figure 2b, with an increase of 0.5. (**a**) Application of the traffic light system to the temporal variation in *b*-value determined using MAXC. The black line represents the temporal variation in *b*-value. The red horizontal dashed line represents the median background level of *b*-values before the M4.7 foreshock occurred. The red, yellow, and green shading correspond to *b*-values decreasing by more than 10%, increasing by more than 10%, and changing by −10~+10%. (**b**) The probability of significance. The gray line is the percentage of ΔAIC>2. The blue line is the percentage of ΔAIC>2 and Δb<0. The red horizontal dashed line is the mean and five times the standard deviation of the background P(ΔAIC>2,Δb<0). (**c**) Temporal variation in D. The red dashed lines through (**a**–**c**) indicate the time-horizon of the foreshock.

**Figure 9 entropy-25-00759-f009:**
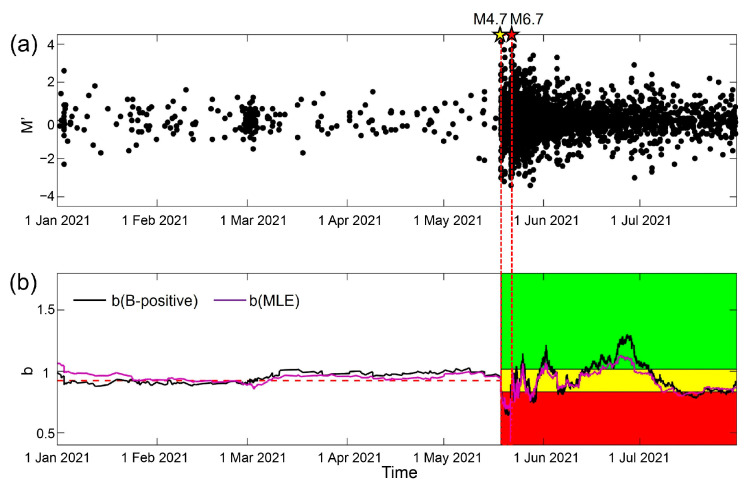
Successive differences in magnitude $M^{\prime }$ and temporal *b*-values using the B-positive method. (**a**) Magnitude difference between successive earthquakes ($M^{\prime }$). (**b**) Temporal evolution of *b*-values using the B-positive method and that used in Figure 2c, and the application of the optimized traffic light system to the temporal evolution of *b*-value. The black line is the temporal variation in *b*-values determined using B-positive. The purple line is the temporal variation in *b*-values in Figure 2c. The red horizontal dashed line is the median background level of *b*-values before the M4.7 foreshock occurred. The red, yellow, and green shading represent *b*-values decreasing by more than 10%, increasing by more than 10%, and changing by −10~+10%. The red dashed lines through (**a**,**b**) indicate the time-horizon of the foreshock.

## Data Availability

The catalog in this study is provided by Earthquake Administration of Yunnan Province, China.

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
