# Peer review of "Optimized Traffic Light System with AIC and Application to the 2021 M6.7 Yangbi Earthquake Sequence"

_entropy, 2023, doi:10.3390/e25050759_

Round 1
Reviewer 1 Report
Authors address an important issue, which is related with operational forecasting earthquake (OEF) systems, and particularly, these ones based on the evolution of the b-value.
In this paper, they apply the traffic light system to the b-value evolution. As a novelty, they propose a new method based on the Akaike Information Criterion (AIC) and bootstrap in order to provide a significance probability of the b-value difference. In this way, the proposed approach improves the effectiveness of the basic traffic light system.
Moreover, two methods were used to obtain the spatial b-values: HIST-PPM, which provides high spatial resolution results; and grid search method that allows verifying the results' reliability.
All the methods used are well described.
The results and discussion sections are also well presented and show the good performance of the proposed method in the Yangbi earthquake sequence.
Besides, the effect of the quality of the earthquake catalog used is also analyzed. Two catalogs are compared, as well as the catalog generated by linear regression of one of them. As a conclusion, a high-resolution seismic catalog can help in estimating the risk after the mainshock.
The results and conclusions are consistent with the main questions addressed.
For all these reasons, I think that the manuscript is suitable for publication.
In line 219, I suppose that you mean ?â„Ž?e?â„Ž??? instead of ?â„Ž???â„Ž???.
In lines 96, 162, 185, 229, you should add a space after the word Figure.
I find the manuscript suitable for publication in its current form.
Reviewer 2 Report
The manuscript is very interesting and well presented, however it should be improved.
My fundamental observation is that the maximum curvature method for the estimation of the completeness magnitude tends to underestimate Mc compromising the correct estimation of the b value. Indeed, for example Lombardi [1] questioned the results of Gulia et al. on the basis of a more accurate estimation of Mc. It is very clear that the Mc value in the red window is smaller than the Mc in the yellow window. This surely reflects in a smaller b value.
My recommendations are to verify the reliability of the Mc estimation in the red window and the adoption of a more accurate method for the estimation of Mc
[1] Lombardi, A. M. (2021). A normalized distance test for co-determining the completeness magnitude and b-value of earthquake catalogs. Journal of Geophysical Research: Solid Earth, 126, e2020JB021242. https://doi. org/10.1029/2020JB021242
